# Quantitative Monitoring of Selected Groups of Parasites in Domestic Ruminants: A Comparative Review

**DOI:** 10.3390/pathogens10091173

**Published:** 2021-09-11

**Authors:** Anna Maurizio, Antonio Frangipane di Regalbono, Rudi Cassini

**Affiliations:** Department of Animal Medicine, Production and Health, University of Padova, Viale dell’Università, 16-35020 Legnaro, Italy; anna.maurizio@unipd.it (A.M.); antonio.frangipane@unipd.it (A.F.d.R.)

**Keywords:** parasite burden, sampling, threshold, endoparasites, lice, mange mites, ruminants

## Abstract

Parasites have had a significant impact on domestic ruminant health and production for a long time, but the emerging threat of drug resistance urgently requires an improved approach to parasite monitoring and control activities. The study reviewed the international literature to analyze the different proposals for the sampling approach and the quantitative estimation of parasite burdens in groups of animals. Moreover, the use of thresholds to decide when and which animal to treat was also investigated. The findings of the study highlighted the presence of a wide-ranging literature on quantitative monitoring for gastrointestinal nematodes (GIN), while more limited data were found for coccidia, and no specific indications were reported for tapeworms. Concerning liver flukes, bronchopulmonary nematodes (BPN) and permanent ectoparasites (lice and mange mites), the diagnostic process is usually aimed at the detection of the parasite rather than at the burden estimation. The main research gaps that need further investigation were also highlighted. For some groups of parasites (e.g., GIN and coccidia) the quantitative approach requires an improved standardization, while its usefulness needs to be confirmed for others (e.g., BPN and lice). The development of practical guidelines for monitoring is also encouraged.

## 1. Introduction

Parasites threaten animal welfare and the economic profitability of many ruminant farming systems worldwide [1,2]. Additionally, the occurrence of acaricide, insecticide and anthelmintic resistance (AR) is increasing at an alarming pace [3]. It is still not clear whether the treatment of ectoparasites with endectocides (i.e., macrocyclic lactones) is an effective driver of anthelmintic resistance for endoparasites [4], but some preliminary evidence already exists the other way round [5]. Hence, an holistic approach to parasite control, addressing both endoparasites and ectoparasites, is of pivotal importance [6]. 

The control of parasites in ruminants generally relies on group-based diagnostics, since individual analyses and selective treatments are still logistically complex in many cases [7,8]. The group-based diagnostic approach should allow for a sufficiently precise determination of the parasitic burden at the herd/flock level, and possibly differentiate among sub-groups (e.g., young, first-year grazing animals, adult) [9] at the same time. Moreover, field veterinarians need a tool to guide them in the interpretation of diagnostic quantitative results to decide when and which groups of animals to treat. In ruminant health management, the presence of parasites does not necessarily imply the need for treatment [10,11], depending also on the group/species of parasite involved. Concerning endoparasites, the scientific community promoted targeted selective treatments (TST) as the ideal drenching strategy to tackle AR, but targeted treatments (TT) are likely to remain the main control strategy for endoparasites in the near future [8] in many ruminant production systems. In this approach, the whole-group, or pre-defined sub-groups of the heard/flock, should be treated only when their parasitic burden is considered at risk. However, an internationally recognized standard approach for the determination of the parasite burden in a group of animals is lacking both for endoparasites and ectoparasites. Moreover, each parasitic group has peculiar characteristics, from both biological and epidemiological point of views, and therefore, the approach to their quantitative determination and for the decision on the treatment holds important differences. 

The present study was based on a literature review aimed at the comparative analysis of the characteristics of quantitative monitoring of different groups of endoparasites and ectoparasites in domestic ruminants. We focused on three aspects of the host-based monitoring process, which are (1) the sampling approach, (2) the quantitative estimation and (3) the threshold determination.

## 2. Methods

A literature review was conducted to critically analyze the differences in the quantitative monitoring approach among selected groups of endoparasites and ectoparasites, including the more relevant, widespread and economically important parasites. 

Among endoparasites, we included the following parasites of the digestive and bronchopulmonary systems: Coccidia, gastrointestinal nematodes (GIN), Trematoda and Cestoda for the former and *Dictyocaulus* spp., and protostrongylids for the latter. Among GIN, gastrointestinal strongyles (GIS) are undoubtedly considered of primary importance [12]. Other endoparasites (e.g., *Schistosoma* spp.), not strictly affecting the above-mentioned systems or not detectable at the copromicroscopic exam, were excluded. Concerning ectoparasites, only the permanent ones (lice, mange mites) were included in the study. Instead, temporary (mosquitoes, flies, sandflies) and periodic (ticks, dipterans causing myasis, fleas) ectoparasites were excluded from our research, in consideration of the important role of the environment for their survival and transmission. In fact, although some of them can be counted during their presence on the host, their host-based quantitative monitoring and consequent strategic treatment do not seem to be applicable.

For the field/laboratory quantitative estimation, we decided to include only parasitological methods that are based on the direct identification and count of adult, immature or reproductive stages of parasites and applicable in alive animals. Indeed, these methods are still the more widespread and sufficiently homogeneous approaches in most of routine diagnostic laboratories.

The study focused on the following domestic ruminant species: cattle, goats, sheep.

## 3. Findings

Several studies have investigated the topic of the quantitative monitoring for endoparasites, and many of the published studies were methodological articles. On the contrary, for ectoparasites, methodological articles were very limited; however, efficacy trials and cross-sectional surveys provided some useful insights. The findings of this review are presented following the three main aspects defined in its aim: (1) the sampling approach, (2) the quantitative estimation methodology and (3) the threshold definition. Concerning the sampling approach, both the number of individuals to be sampled in a herd/flock and the collection of individual versus pooled (or composite) samples were considered.

### 3.1. Sampling Approach 

#### 3.1.1. Sample Size Determination 

The sample size determination has been scarcely discussed in the literature for both endoparasites and ectoparasites. This information for ectoparasites is almost completely lacking, but some general indications can be extrapolated from cross-sectional surveys. A survey of cattle lice in dairy herds of New York [13] was carried out, sampling 10 mature cows and 10 calves per farm. In another study [14] comparing three methods (hair partings, lamp test and table locks test) of detection of lice in sheep, a sample size of 10 animals per flock was used and whenever the sheep with clinical symptoms were present, these were included among the 10 sampled animals. Inspecting by hair partings in 10 instead of 5 sheep and on both sides of the animal instead of one increased the test sensitivity from 0.46 to 0.71 (P = 0.08), but James et al. [15] suggest that increasing the number of sheep increases the sensitivity more than increasing the number of parts per sheep by an equivalent amount. 

The sample size (*n*) suggested to monitor endoparasites in a flock/herd usually followed generic indications, with 10 (ranging from 7 to 20) animals per farm [16,17,18,19,20,21,22,23] or 10% of the flock [20,24] being the most frequent recommendations. For coccidia, the sampling sizes suggested were generic per flock (7–10 animals) or specific for young animals (i.e., 6–16 young animals) [25,26]. Sampling a percentage of the flock allows the sample size to be adjusted to the farm size (*N*). However, Hansen and Perry [27] pointed out that the amount of animals to be sampled should not necessarily be increased/decreased proportionally to the farm size. Their suggestion on sample size determination, based on both general indications and on practical/logistical constraints, is shown in Figure 1. The non-proportional increment is clearly demonstrated in the same figure by a more recent study carried out on goat farms [28], which is proposing a statistical determination of the sample size. 

Specific indications regarding which individuals to sample can be present, as different animal categories can show a different susceptibility to parasites or be specifically targeted for treatment. For instance, symptomatic [27] and asymptomatic [18] animals or specific age classes (e.g., calves or young adults in grazing herds/flocks [19,28]) may be preferentially sampled. For coccidiosis, some authors [25,26] recommend only sampling calves/kids.

The monitoring of liver flukes and BPN usually focused on the detection of the infection rather than on the burden determination and protocols using parasitological methods on alive animals are very limited. A composite *Fasciola hepatica* fecal egg sedimentation test was recently developed [29] for cattle: a mixture of adult suckler cows and their calves were sampled, for a total of 20 animals per farm. In sheep, ten pools of six samples each per flock were indicated as a cost-effective protocol for the diagnosis of protostrongylid infections [30]. The sensitivity of the test is affected by the within flock prevalence, but also by the age of the animals sampled. Indeed, it was seen in cattle that sampling heifers requires fewer animals to detect the *Dictyocaulus viviparus* infection compared to sampling cows or the entire herd [31]. In this latter study, the sample size for the detection of the infection was also statistically calculated for each farm involved.

#### 3.1.2. Pools vs. Individual Samples

The sampling approach can be based either on individual or on pooled samples. This is usually not the case for ectoparasites, whose evaluation tend to be more individual. The use of composite samples was instead widely investigated for endoparasites, and mostly taking into account the gastrointestinal nematodes. Individual and composite samples both have their advantages, such as lower costs and labor for pools and a more complete portray of the infection burden for individual samples. As such, some authors recommend the use of individual samples [20,28], while others prefer pooled samples as more applicable in common practice [16,32]. The literature concerning the use of composite samples is wide and several methodological studies exist. Table 1 summarizes the number of samples to be included in each pool, as proposed by different authors. The pool sizes ranged from 5 to 20 in cows and from 3 to 20 in sheep, while 3–8 samples were used in the only study performed on goats. The concordance or agreement with the corresponding individual samples was evaluated with different statistical approaches, but it was generally very high. Nevertheless, a study carried out in sheep for GIS showed that as the egg output increased, the FEC based on pooled samples provided lower estimates compared to the mean FEC from the corresponding individual samples [32]. The opposite situation was instead recorded in goats, where the pools tended to overestimate the abundance of GIS in the case of a high FEC, and to underestimate it for lower egg outputs [33]. Clearly, the correlation can be impacted if the contribution of individual animals to the pool is unequal.When monitoring coccidia to obtain a good estimate of the burden of coccidiosis within a group of calves, an examination of fecal samples from several individuals is recommended. The samples may also be pooled [34], but clinically ill individuals should be sampled separately to estimate the amount of pathogenic *Eimeria* oocysts excreted [35]. 

Finally, one protocol each based on the pooled samples was found for Fasciola hepatica [29] and protostrongylids [30], respectively, for cattle and sheep, although the aim was the detection of the infection and not the burden estimation. The pool size used for protostrongylids (6) was not reported in Table 1 as it was calculated based on the specific expected prevalence of that study.

### 3.2. Quantitative Estimation Methodology

The quantification of endoparasites is a common practice in ruminant farming and it is traditionally carried out using FEC, where parasitic elements (i.e., eggs, oocysts and larvae) are counted. However, while the use of FEC is widespread and can be considered a standard approach for quantitative monitoring of endoparasites, the specific techniques still lack standardization [39]. Besides the multitude of published protocols, laboratories and research facilities often apply further modifications, particularly pertaining the amounts of feces analyzed, the sample dilution, the flotation solution employed and the centrifugation settings [39,40]. In addition, the entity of the correlation between the number of parasitic elements recovered per gram of feces and the number of adults in the host remains unclear. Evidence supporting this correlation exists for small ruminants [41,42], while the correlation is considered poor in cattle [43], with a possible exception for first-time grazing calves [27]. The McMaster technique and its modified versions remain the most widely used techniques in veterinary parasitology [40] but newly developed techniques, i.e., the FLOTAC (which was also successfully validated for the diagnosis of gastrointestinal protozoa, liver flukes and BPN) and its evolution, the mini-FLOTAC, seem promising. Indeed, these have higher sensitivity compared to the McMaster method, and standardized protocols were published [44,45]. The main limit of FEC techniques is the inability to distinguish between eggs of different genera/species of strongylids. Larval culture (LC) can be used to overcome this limit, but this procedure carries major operational disadvantages (e.g., time and expertise required, risk of bias due to the culture conditions, no automation), and current efforts by the scientific community are directed to the substitution of LC with molecular techniques. Molecular tools can overcome many of the limits of LC and their higher sensitivity and specificity better respond to the challenges that AR is posing on the livestock sector. In addition to qualitative PCR techniques, the development of quantitative PCR (qPCR) assays is attracting the attention of many researchers. After the first efforts of von Samson-Himmelstjerna et al. [46] over twenty years ago, there have been significant advances in the field. Few assays for the quantification or semi-quantification of nematode eggs from sheep [47,48,49] and cattle [50] feces were published and even tested in field conditions, with promising results [51]. More recently, a single multiplexed qPCR assay [52] was also developed to both identify and quantify *Haemonchus*, *Trichostrongylus* and *Teladorsagia* eggs in sheep feces. While the further development and validation of this technique are still required, it holds the potential to substantially reduce the time and costs of the diagnosis, thus allowing its routine use for monitoring and epidemiological purposes.

FEC techniques are also used, besides GIN, to quantify the coccidia oocysts per gram of feces (OPG). The information provided is not very reliable as the excretion levels may vary greatly over time in positive individuals [35] and clinically healthy animals can excrete thousands of OPG, while in others, acute coccidiosis may occur before oocysts are demonstrable in feces. However, when clinical symptoms are present in animals with a high oocyst output, this can be highly suggestive of the disease [26]. Moreover, many *Eimeria* species are not or only mildly pathogenic; therefore, the determination of the species is crucial in the diagnosis [21,35]. With traditional coprological methods, species identification is very labor-intensive, requires skillful personnel and takes a long time (due to the use of coproculture), and it is not always possible to obtain an accurate discrimination of the species; thus, also for coccidia, molecular tools will probably be implemented in the future [26].

Concerning liver flukes, the presence of eggs in the feces is still considered the best way to confirm the infection in an alive host, even though copromicroscopical analyses are of limited value in newly infected animals. Flotation techniques using high density solutions can be used but the shape of eggs may be affected; therefore, sedimentation methods are recommended [53]. Diagnostic tests for *Fasciola* usually focus on the sole presence or absence of the infection, but knowledge of the intensity of infection should be promoted [54,55], as it would support a more sound use of anthelmintics. The correlation between FEC and parasite burden has been generally considered weak, also due to the intermittent shedding of parasite eggs following storage in the gall bladder. However, a strong positive correlation was reported in a recent study [55]. 

The infection by BPN is usually assessed using the Baermann technique and its modifications to recover first-stage larvae from the feces. As for the liver flukes, this evaluation is generally only qualitative, although it can be adapted to quantify the larvae per gram of feces (LPG). The FLOTAC method was also validated for the diagnosis of BPN of ruminants. Compared to the Baermann technique, FLOTAC cuts down the time for sample processing and provides higher counts of LPG [44]. However, when tested on bighorn sheep, FLOTAC’s performance in detecting mixed infections was poorer compared to the Baermann technique and the authors suggested that FLOTAC may be more useful in studies where total counts, and not taxonomic identity, are the primary objectives [56]. A quantitative method for the count of BPN larvae, based on a modified Baermann technique, was developed in a study on wild ruminants [57], and can be easily adapted to domestic ones.

Methodological studies on how to quantitatively monitor the ectoparasites in ruminants are lacking and the focus is generally on the detection of the parasites, rather than on the actual quantification of the burden. The techniques commonly used in the literature are visual examination and hair partings for lice, and scrapings from the periphery of lesions for mange. If lesions regressed during the study, scrapings from the area of previously active lesions could be considered [58]. An alternative approach was recently established by Cassini et al. [59] in a efficacy trial for two ectoparasiticides in cattle. Following a preliminary observation, the dorso-lateral surface of the animal was sheared in three different areas and the hair subsequently collected to be examined under a stereomicroscope for lice identification and counting. In another study [14], the lamp test, which uses the negative phototaxis of lice to separate them from the sheared wool, and the table locks test, where sheared wool is dissolved in sodium hydroxide heated in a water bath and the residue subsequently examined, were used in parallel with hair partings. Monitoring approaches for ectoparasites in the literature are very heterogenous, with a multitude of protocols varying for the area or length of skin surface examined and for the position and number of sites considered (Table 2). 

The examined sites were most commonly located along the dorsal line of the animal, particularly around the neck, the shoulders and the rump, but the brisket in cattle [60,61,62,63] and the flanks in sheep [14,15,64] were also frequently included. Differently, the World Association for the Advancement of Veterinary Parasitology (WAAVP), in the guidelines for evaluating the efficacy of ectoparasiticides against the lice of ruminants [65], indicated the determination of the sites to be included through a preliminary examination in cattle, and to select sites representative of the full area of body covered by fleece in sheep. Finally, while most of the protocols aimed to detect lice or mites in general, some focused specifically on a single species of lice (i.e., *Bovicola ovis* [14,15], *Bovicola caprae* [64] and *Linognathus vituli* [62]) and mites (i.e., *Psoroptes ovis* [3] and *Chorioptes bovis* [58]). 

### 3.3. Threshold Definition

The use of thresholds for gastrointestinal nematode control, and especially for GIS, is of common practice and the thresholds discussed in the literature focused on these parasitic groups. Although many factors can influence the impact of a parasitic infection on the host, the definition of thresholds may support the decision on whether or not to treat individuals (targeted selective treatment, TST) or groups of animals (targeted treatment, TT). Furthermore, few guidelines can be found in the literature to aid the interpretation of low, medium and high FECs based on the genera of parasites present in adult sheep [18] or in young animals (cattle and sheep) [6,27]. The thresholds proposed in the literature were summarized in Table 3; only those based on parasitological analyses were considered. The thresholds adopted as a basis for TST in practical surveys were also included. In cattle, treatment was indicated for generally lower EPG (200 EPG) compared to sheep and goats (200–2000 EPG). Several authors underline the importance of the identification of the genera/species involved, in particular when highly prolific genera (e.g., *Haemonchus* or *Oesophagostomum*) are involved [2,7,20,39]. 

A clear threshold for coccidia is difficult to assess [21,26], since the pathogenicity differs among *Eimeria* species and the correlation of the infection intensity with the clinical conditions is poor [34]. In sheep, 250–500 OPG of the very pathogenic species may have the same impact of 10,000 OPG of other species [25]. Yvorè et al. [78] point out that clinical coccidiosis would unveil whatever the *Eimeria* species involved whenever the oocyst output exceeds 50,000–100,000 OPG, but this cannot be used as a threshold for treatment. Further reasoning is possible if we achieve a reliable identification of the *Eimeria* species involved. In general, the mere presence of highly pathogenic *Eimeria* species in a group of young animals is considered a potential threat to animal health [35]. About 500 OPG of pathogenic *Eimeria* are considerate indicative of a threat to the individual health of cattle, while no threshold has been established thus far in sheep and goats [35], even though, on a group level, a threshold of 500 OPG in sheep was recently proposed as indicative of an active infection [25]. 

Concerning liver flukes, Rojo-Vàzquez et al. [53] suggest that, in their experience, an output of 100 to 200 EPG is indicative of an active and severe infection, which requires the use of a flukicide. However, on a productive level, it was seen that the presence of only 1–10 flukes affects the growth rate of beef cattle [54,55]; therefore, it could be somehow inferred that the recovery of eggs in faces is already indicative that the production is impacted. Furthermore, even qualitative sedimentation-flotation techniques only detect the most heavily infected animals, indirectly providing information about the infection intensity. Indeed, in the effort of identifying a treatment threshold for flukes, Vercruysse and Claerebout [22] calculated a theoretical threshold of approximately 90 EPG as indicative that treatment is necessary. However, they highlight that in most cases, even in heavily infected herds, egg outputs are much lower, commonly less than five EPG. As such, 5 EPG was recommended as an acceptable threshold estimate.

No thresholds were proposed for BPN.

Equally, no thresholds were proposed for ectoparasites either, but a rating scale for lice [63] and density scores for both lice [61,62] and mites [3] were defined in cattle. These systems had the purpose of semi-quantitatively defining the infection intensity, but no consideration was made on the possible use of these scales/scores in defining a treatment threshold.

## 4. Discussion

Endoparasites and ectoparasites share some similarities in their biological characteristics, but they have important differences, too. One general aspect of parasites is their distribution in the host population: parasites tend to be aggregated across the host population, with the majority of the parasite population concentrated into a minority of the host population [79]. Thus, the few animals in a herd/flock that harbor high burdens of parasites are more prone to have clinical or sub-clinical consequences and, moreover, they are responsible for most of the parasite transmission and persistence. The identification and treatment of these animals should be one of the main targets of a monitoring and control activity. Moreover, the aggregated distribution of parasites can complicate the interpretation of a sample-based quantitative assessment aimed at the estimation of the overall burden in a group of animals, as is usually conducted for ruminants. The aggregate distribution was widely investigated in the literature and is clearly demonstrated for endoparasites, mostly thanks to ecological studies in wild host populations [80], but the tendency towards aggregation is also certainly present in domestic ruminants [81]. Concerning permanent ectoparasites (lice and mange mites), there are less studies on their distribution patterns by far, but their tendency to aggregate has been demonstrated already [66]. 

An important difference between the two groups of parasites is related to their life cycle. All the endoparasites considered in this study have an environmental phase; they reproduce inside the definitive host (ruminants) and their transmission to other hosts is highly affected by chance. The coccidia life cycle is quite rapid (2–4 weeks), whereas the life cycle of helminths requires time for the development of the external and internal stages, resulting in long durations (1–6 months). On the contrary, ectoparasites living permanently on the host are transmitted by direct contact and their life is characterized by many reproduction cycles in a relatively shorter time (2–4 weeks) [82]. These biological characteristics have important consequences on parasite dynamics and, therefore, on their monitoring. The fluctuation of the helminths burden is a long-term one, while the burden and the severity of a coccida infection are linked to a self-limiting behavior [82]. Indeed, endoparasite populations tend to reach an ecological equilibrium with their host population, and the modification of this status generally needs a longer period of time. On the contrary, the fluctuations can be very fast for a permanent ectoparasite, partly due to the higher efficacy of their direct transmission, which allows lice and mites to spread rapidly in a group of gregarious animals, such as ruminants. These differences between ectoparasites and endoparasites are reflected in the different approaches proposed by the scientific literature for their monitoring, as highlighted by the findings of our study, which focused on the sampling approach, the quantitative estimation and the use of thresholds for a treatment decision. These findings and the research gaps are discussed in the following paragraphs.

### 4.1. Sampling Approach

Most of the literature about the sampling approach dealt with endoparasites, and only for this group, the use of composite samples was investigated. Probably, due to the spread of AR, the need for selective and timely treatment interventions became urgent for endoparasites, calling for an approach that should be precise and cost-effective at the same time. The use of TST represents the final goal in this field, but farmers are still reluctant to abandon traditional treatment schemes [7]. However, as long as the efficacy of anthelmintics is constantly monitored, the use of TT is still acceptable in areas where AR is rare [8]. Therefore, although TST should be strongly promoted in any farming context, efforts should also be addressed to ensure the best targeting of whole-group treatments. Consequently, pending issues concerning group-level diagnostics, such as the sampling size and the use of composite sample, should still be tackled. This also applies to ectoparasites, although less discussed in the literature. Interestingly, albeit the control of ectoparasites generally also has a whole-group basis, indications on the sample size for monitoring are completely lacking. The lack of information on the sampling approach, not only for ectoparasites but also for liver flukes and BPN, can be partially ascribed to the fact that their monitoring usually aims at the detection and not at the quantification of the infection. Nonetheless, this should still be highlighted as a research gap that requires further investigation. Nevertheless, some aspects for GIN were also quite neglected in the literature, for which the quantitative approach is widely adopted. The indications for sample size remained very generic (i.e., 10 animals per farm) and [28] a formula to optimize the sample size, taking into account the aggregation level, was developed only recently for goats. Moreover, this formula allows the sample size to be tailored to the farm size, providing a relevant step forward in the statistical definition of group-based diagnostics in ruminant farming. Aggregation has implications for the diagnosis of the endoparasite burden in a group of animals: with very high levels of aggregation, a small sample size is likely to underestimate the actual burden [83] and this tendency becomes more marked as aggregation increases. Ideally, the sample size should be adjusted to the degree of aggregation and to the expected intensity of infection [17]. A similar situation has also been suggested for ectoparasites [10]. These aspects call for further research toward the development of a commonly accepted statistically based method for the determination of the correct sample size for the estimation of the parasite burden in a group of animals.

The use of composite samples was only studied for endoparasites, and it has been seen that pooled FECs generally have very high levels of correlation/agreement with the mean FECs from the corresponding individual samples. Several methodological studies confirmed this result [16,17,32,33,36,37], regardless of the number of samples included in the pool (in a range from 3 to 20). This finding, coupled with the reduction in costs and processing time of the analysis, makes the use of composite samples very convenient in common practice, not only for GIN but also for coccidia, even though, for the latter, methodological studies investigating the correlation/agreement with individual samples are lacking. However, the FECs obtained from pooled samples lack a confidence interval that can provide significant information on the level of parasite aggregation (high ratio variance/average suggest high aggregation level). The information on the variability of the sampled animals among FECs is the main advantage of using individual over pooled samples. In some cases, in fact, an individual approach to diagnosis may optimize parasite control practices in the farms. If opting for the use of composite samples, a clear indication on the number of samples to include in a pool was not provided by the literature, since the above-mentioned agreement/correlation was confirmed, regardless of the pool size. Anyhow, a defined pool size would be useful to standardize cross-sectional surveys, while it would probably be better to adapt the pool size to the managerial characteristics of each farm during in-farm monitoring.

What needs further investigation is which animals to sample. On one hand, for a whole group approach, the sample should be representative of the flock/herd. On the other hand, specific host categories may be preferentially sampled (e.g., young animals for coccidia, first year-grazing animals for strongyles) to assess the infection burden in the individuals more at risk. This could potentially lead to an intermediate step between TT and TST, which is the subgroup treatment [9]. However, this approach needs to be based on sound field-collected data and tailored for the different geographical areas and for each specific farming system.

### 4.2. Quantitative Estimation

The quantitative estimation of the parasite burden is a common procedure for gastrointestinal parasites and FEC techniques are a conventional diagnostic approach in ruminant farms. These techniques have the following several advantages: they are user-friendly, low-cost and do not require specialized instrumentation. However, FEC methods lack standardization and have a limited sensitivity (10–50 EPG for the McMaster technique [39]). The outcome of the analysis can also be influenced by several operational (e.g., storage conditions), physical (e.g., fecal dry matter) and biological (e.g., seasonality, host immunity and reproductive status) factors. The development of FLOTAC techniques increased the sensitivity of FEC up of 1 EPG and represented a promising step forward, especially in the diagnosis of low infection levels, which are, for instance, very common in cattle [39]. Moreover, a standard processing procedure was described in-depth for FLOTAC techniques [45,84]. Nonetheless, it is critical that, regardless of the technique employed, the interpretation of results remain flexible, with additional care when pooled samples are used [23]. Indeed, in addition to the above-mentioned factors, the composition of the parasite population in the host also has great implications for the significance of the egg output. First, prolificity varies greatly among parasite species. Many uncertainties remain about the correlation between FEC and worm burden, but the correlation is considered lower for genera with a low prolificity (i.e., *Trichostrongylus* and *Teladorsagia*) [39]. In second place, the genera/species of both helminths and coccidia significantly differ in their pathogenicity and, therefore, in their clinical relevance, conditioning the treatment decision. The discrimination of the genera/species involved is not possible with FEC techniques, and LC, although useful, holds too many operational limits to be employed as a standard diagnostic procedure. Further reflections are needed about the interpretation of FEC, but molecular tools are expected to play a main role in the future, overcoming many of the limits of the traditional techniques [39]. The development of quantitative assays may even bring the complete substitution of FECs with molecular techniques, although this is not expected as a short-term achievement. 

Contrary to gastrointestinal parasites, the monitoring approach is much less standardized for ectoparasites and, with the exception of the guidelines for efficacy studies [60,65], no clear indications were published for farm monitoring. Usually, mites are searched with scrapings around the lesions, while lice are counted directly on the animal on multiple sites, following the parting of the coat or not. However, as Table 2 shows, the location, area/length and the number of sites examined in the literature are extremely heterogeneous. The monitoring of ectoparasites is generally aimed at the detection and not at the quantification of the infection; therefore, the lack of a standardized approach for quantitative monitoring can likely be attributed to this. Some ectoparasites, such as chorioptic and psoroptic mites, represent an important threat to the welfare of their host and usually there is a high likelihood of a rapid and predictable increase leading to clinical disease [10] and spread of the infection [85], due to their above-mentioned biological characteristics. However, in some cases, such as for lice [10,86], the presence of a low infection burden can likely be well tolerated by the host. Therefore, standardized methods to quantify the burden and clear indications for farmers and veterinary practitioners should be available. In most of the analyzed papers, the quantitative approach was used to study the effect of infestation or for drug efficacy trials, while no specific methodological paper investigated the use of this approach for in-farm monitoring. However, if appropriately adapted to the field conditions and adequately standardized, these quantitative methods are probably allowing for a more sensitive diagnostic approach and consequently assure a prompter intervention.

### 4.3. Threshold Definition

It is acknowledged that the presence of a certain burden of parasites in ruminants is common and the complete elimination of the infection would not be feasible [22]. Hence, farmers and practitioners tend to rely on thresholds for treatment decisions, at least for what concerns GIN and coccidia. SubSection 3.2 briefly highlighted how many factors influence FECs and, as such, it is clear that the establishment of one univocal threshold is more than unrealistic [22], especially given the complexity of each parasite–host–environment relationship. With regard to coccidia, the enormous differences in the pathogenicity and oocysts output among species hindered the debate over treatment thresholds. The identification of the *Eimeria* species involved is now considered a prerequisite for the definition of thresholds by several authors [21,35]. Evidently, the development and implementation of molecular assays in standard diagnostics is expected to provide the tools for the use of thresholds for coccidia in common practice. In the meantime, however, until these tools are widely available, researchers should address some efforts on the topic. Indeed, if the establishment of strict thresholds is unrealistic, and many objections can be raised against their use, their role to promote a more sustainable use of antiparasitic drugs is undeniable [22]. For GIN, the literature is wider, and several thresholds were proposed (see Table 3). The authors agree over the value for cattle, which was consistently 200 EPG. For small ruminants, suggested thresholds were higher compared to cows, as a consequence of the higher burdens found in these species, but with a great variability (200–2000 EPG). This variability highlights the difficulties related to the definition of a threshold. Moreover, it has been seen in cattle that reasonably accurate thresholds can be determined for different age classes [22], moving the focus from host species to specific sub-categories. Another option may be the study of thresholds tailored on the composition of the parasite population. Even if whichever threshold should not be strictly used, these alternatives could help to improve the fitness of FEC-based thresholds as treatment indicators and represent an interesting matter for future research. The assessment of productive or pathophysiological parameters (e.g., FAMACHA), which are of growing interest as low-cost user-friendly treatment indicators, could be coupled with FECs for a sounder decision [8].

In the previous subsection, it was discussed how the presence of some ectoparasites is simply not tolerated [10,86]. Hence, the approach to their monitoring is usually only qualitative. It is only for lice that low parasite burdens are considered acceptable [10], opening questions about which threshold to use. The literature is very limited over this point. No thresholds were proposed for lice but rating scales [63] and density scores [61,62] were described in cattle; therefore, it could be investigated whether these can be used as a basis to determine a treatment threshold. Indeed, individual thresholds should clearly be established in the future, since the implementation of TST was indicated as a practice for lice control [86]. The ability to selectively treat only the more susceptible and heavily infected individuals is expected to reduce the number of treatments with consequent lower costs for the farmer and a reduced risk of resistance onset [10]. However, further evaluations are needed in the future, since the impact of these parasites on the host’s welfare is not completely understood. Moreover, the potential risk of a high rate of increase and spread should also be considered for ectoparasites when deciding for or against treatment [10]. Hence, risk assessment tools should be developed for use in common practice.

### 4.4. Bronchopulmonary Nematodes, Tapeworms and Liver Flukes

The approach to BPN, tapeworms and liver flukes is substantially qualitative. As a result, the literature about their quantitative monitoring is extremely limited. In first place, no clear indications exist over the sampling approach. It is only reported that the number of samples to be examined should be as large as possible, especially for older animals [31], when trying to detect BPN in a herd [22]. The use of pools was evaluated only in one study each for *Fasciola hepatica* [29] and protostrongylids [30], respectively for cattle and sheep, but never with a quantitative objective, even though quantitative diagnostic methods exist for both parasite groups. In addition to the standard diagnostic techniques (FEC for liver flukes and Baermann for BPN), FLOTAC was also recently validated for both liver flukes and bronchopulmonary nematodes [44,84], but further field testing is needed for a better comparison between these techniques [56]. For liver flukes, FEC remains the main diagnostic technique, but several other techniques are available in parallel and will probably be increasingly used in the future. Among these, the coproantigen ELISA seems to have the best performance in indicating the level of infection [54,55]. The establishment of a treatment threshold, however, is still difficult, due to the lack of data from natural infections [22]. There is no agreement over the few threshold indications for *Fasciola* [53,54,55], but they clearly indicate that treatment is needed for very low emission levels. Moreover, Mazeri et al. [55] pointed out that even only 1–10 flukes in the liver can decrease the productivity in beef cattle. Clearly the use of thresholds for liver flukes requires further study, since the selective use of flukicides is expected to have an increasing importance in the future [87]. No thresholds were suggested for BPN. In one study [71], which investigated the use of TST to control nematodes in dairy calves, treatment was performed for FEC>200 EPG or at the detection of BPN. The authors explained this choice with the lack of guidelines on the treatment of calves subclinically infected with lungworms and with ethical concerns over the welfare of the calves. Both these topics deserve further investigation, and research should also be carried out in older cows and other species. Additionally, the importance of a careful risk assessment was highlighted whenever deciding if treatment against BPN is necessary [22]. A fluke forecast is available for certain regions and has proven its usefulness [22]. The development of a similar tool would likely also be useful for BPN.

## 5. Conclusions

The findings of our review allowed for a comparison between the monitoring and control approaches proposed in the literature for selected groups of parasites. The main research gaps, and the consequential way forward in future research, were identified and conceptualized. 

The study highlighted the presence of many investigations related to the quantitative monitoring of coccidian and GIN infections. Some indications on the correct number of animals to be sampled were provided for these parasitic groups, together with the identification of thresholds, usually in terms of oocysts/eggs output, that were mostly suggested for promoting their use in TST. Composite samples were proposed as a more cost-efficient alternative to individual samples, in consideration of the high agreement found between the two approaches in most studies. However, there are still some aspects of the monitoring approach that deserve further investigation and stronger agreement among researchers, such as the more appropriate approach for sample size determination and for the use of thresholds. The use of the mean OPG/EPG in a group or that of a composite sample OPG/EPG as a threshold for TT is a research issue that is basically still unexplored.

Among endoparasites, liver flukes, tapeworms and BPN were rarely or never investigated with a quantitative approach and many authors agreed that for some of these parasites (e.g., *Fasciola hepatica*), the detection of the reproductive stages should already be suggestive of an appropriate treatment, in consideration of their high pathogenic potential. However, an improved understanding of the correct approach to quantify the first-stage larval output of BPN and also the identification of potential thresholds for these parasites are research topics that deserve more attention.

The diagnostic approach for permanent ectoparasites (lice and mange mites) is mainly aimed at the detection of the parasite rather than at its burden estimation, and this is probably due to their capacity to rapidly spread in a herd/flock and to affect the health status and the productivity of the hosts. However, in the literature, we found many studies (mainly drug efficacy trials or epidemiological surveys) using a quantitative approach for the determination of parasite burden, although their methods were extremely variable. An improved standardization in the methods used to quantify lice and mites is strongly welcomed.

Finally, our study performed an initial systematic analysis on some specific aspects of the approach used in the field and proposed by experts for the monitoring and control of selected types of parasites. We encourage further discussion among scientists on this topic, using the evidence-based data already available in the literature to appropriately address the need of herders and field veterinarians for clear and practical guidelines.

## Figures and Tables

**Figure 1 pathogens-10-01173-f001:**
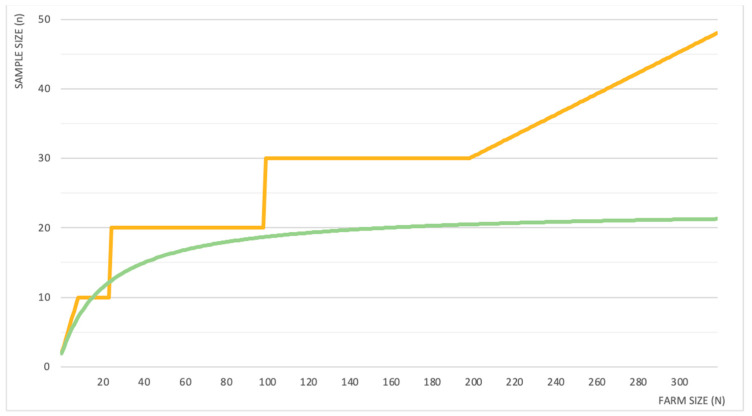
Number of individuals to be sampled (y-axis) in a group of animals (farm) based on its size (x-axis) to obtain a good burden estimation, according to Hansen and Perry [27] (in orange) and Maurizio et al. [28] (in green).

**Table 1 pathogens-10-01173-t001:** Pool sizes used in methodological studies in the literature and relative correlation/agreement with corresponding individual samples. In all studies, equal amounts of feces from each individual were used to compose the pools.

Host Species	Parasite Group	N Samples Pool	Correlation/ Agreement	Statistical Approach	Ref
Cattle	Helminths	5	0.86	Level of agreement (K statistic)	[16]
	GIN	5 (3–6)	0.98	Spearman’s rho correlation coefficient	[36]
		10 (6–10)	0.97		
		global* (9 20)	0.95		
		5 (3 6)	0.99	Lin’s concordance correlation coefficients	
		10 (6 10)	0.97		
		global* (9 20)	0.97		
	*Fasciola hepatica*	10	-	-	[29] ^a^
Sheep	GIS	3 5	0.94	Coefficient of correlation	[37]
			0.80	Level of agreement (K statistic)	
		10	-	-	[38] ^a^
		5 (3 5)	0.94-0.97 ^b^	Pearson correlation coefficient	[32]
		10 (8 10)	0.94-0.99 ^b^		
		20 (17 20)	0.99		
	GIN	10	0.97 (0.90–1.04)	Regression constant (95% CI)	[17]
		3	-	-	[24] ^a^
Goat	GIN	3 8	0.88	Pearson correlation coefficient	[33]

GIN = gastrointestinal nematodes; GIS = gastrointestinal strongyles. * = all sampled animals per farm; ^a^ = the study proposed a monitoring protocol, without investigating correlation/agreement between pooled and individual samples; ^b^ = three copromicroscopic techniques were compared

**Table 2 pathogens-10-01173-t002:** Overview of protocols to quantify lice and mites in the literature: method, number and size of sites examined and total length/area examined on the animal considering all the sites.

Host Species	Parasite Group	Protocol	Ref
Method	Sites	Description	Length/AreaExamined (tot)
Young cattle	Lice	VE	5	10 × 10 cm each	500 cm^2^	[62] ^a^
HP	5	3 partings of 10 cm/site	150 cm	[66] ^b^
Calves	Lice	HP	4	30 cm^2^ each	120 cm^2^	[13]
Cattle	Lice	HP	4	150 cm^2^ in total		[13]
HP	22	5 × 40 mm	44 cm^2^	[61] ^a^
VE	5	6.5 cm ^2^ each	32.5 cm ^2^	[63] ^c^
VE	8	Counts over a predefined surface area/site (7 trials) or along a predefined number of hair partings (3 trials)		[67] ^d,e^
Shearing, VE and microscopeobservation of hair	3	50 cm^2^ each	150 cm^2^	[59] ^d^
HP	6	At least 3 partings of 5–15cm/site	90–270 cm	[65] ^d,f^
Mites	S		≤ 10% of each lesion		[67] ^d^
S	3	4 cm^2^ each	12 cm	[68] ^g^
S and HP	scraping in ≥ 3 lesions			[60] ^d,h^
S	6	3 × 3 cm	54 cm^2^	[58] ^d,i^
S	6	6 cm^2^ each	36 cm^2^	[3] ^a,d^
Sheep	Lice	HP	40	10 cm each5 cm each	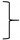	400 cm	[65] ^d,f^
HP	80
HP	5	10 cm each	50 cm	[14]
Lamp test	2/flank	2 g of wool/each		[14]
Table locks test	4	60 g of locks wool in total		[14]
HP	20/69			[15]
Goat	Lice	HP	26	5 cm	130 cm	[69]
HP	20	6.5 cm^2^ each	130 cm^2^	[70]
VE	6	10 × 10 cm each	600 cm^2^	[64] ^d^

Method: visual examination (VE), hair partings (HP), scraping (S). ^a^ = use of a scoring system; ^b^ = count up to 30/parting. If higher: “> 30”; ^c^ = use of a rating score; ^d^ = efficacy study; ^e^ = counts by lice species; ^f^ = WAAVP guidelines; ^g^ = mites counted up to 100 (3 trials) or total living mite counts (4 trials); ^h^ = EMEA indications; ^i^ = count up to 100/scraping, if higher: “> 100”.

**Table 3 pathogens-10-01173-t003:** Thresholds present in the literature for domestic ruminants and their intended use.

Host Species	Parasite Group	Suggested Threshold		Ref
		EPG	Intended Use	
Cattle	Helminths	200	NS	[16] ^a^
Dairy calves	GIS	200	TST	[71]
	GIN	200	NS	[2] ^a^
Sheep	GIN	500–1000 ^b^	NS	[2] ^a^
	GIN	500	TT	[17]
	GIN	500	TST	[72]
	GIS	800	TST	[73]
	GIS	Mean FEC of the group	TST	[74]
	*Fasciola hepatica*	100–200	NS	[53]
	GIS	400–600	TST	[7] ^c^
Sheep, goat	GIN	Ewes: 1000 ^d^ in late spring, 2000 ^e^ at the end of grazing season; goats: 500 ^f^	TT	[20] ^a^
	GIN	300	TST	[75]
	GIN	200	NS	[39] ^c^
	GIN	300	TST	[76]
Goat	GIS	Mean FEC of the group	TST	[40] ^a^
	GIN	500–750 ^g^	TST	[77]

Intended use: individual threshold for TST, average group threshold for TT, not specified (NS). ^a^ = refers to other studies; ^b^ = based on *Haemonchus* absence–presence; ^c^ = highlights the importance of the identification of nematodes; ^d^ = EPG of *Haemonchus contortus*; ^e^ = in non-lactating ewes; ^f^ = in goats at risk of haemonchosis; ^g^ = increased in the second year of the study due to the lack of clinical problems in the first year.

## Data Availability

All data reported in the review can be found in the cited studies.

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
