# Peer review of "Quantitative Monitoring of Selected Groups of Parasites in Domestic Ruminants: A Comparative Review"

_pathogens, 2021, doi:10.3390/pathogens10091173_

Round 1

Reviewer 1 Report

The manuscript entitled "Quantitative monitoring of endo- and ectoparasites of domestic ruminants: a comparative review". Title, abstract and overall rationale of work to some extent is good. However, there are some major concerns, which needs to be addressed and needs substantial revision.

1) Introduction section is too lengthy and I would like to suggest author would be better to represent some information in the form of image(s).

2) I would like to suggest author, they must be include these ectoparasites such as fleas and ticks.

3) I would like to suggest author to show a pictorial view of this (Sample size determination) part because this is too lengthy and not attractive.

4) Author explain about the liver fluke parasite (Fasciola hepatica) and what’s about status of Schistosoma parasite in the animals.

5) A flowchart should be added to the article to show the clear mechanism of this research.

6) The most drawback of this review article are: written part is more, no figure and too much lengthy written, I would highly suggest the author they must include 2-3 figure and also I suggest the reduce the written part and try to express in the picture form.

7)  Author need to summarize main theme of this review and need to explain and highlight about the strength of this review.

8) In this section (Bronchopulmonary nematodes, tapeworms and liver flukes) author explain about the different technique using to determine the parasite line number (525-536) however, there are several modern technique is are available to detect of these parasites live qRT-PCR, ELISA, Gene sequencing and now a day most of the lab using these techniques to identify the parasites.

9) I am totally agree with the author they explain about the threshold definition section and I would like to suggest author also explain the limitation of this section.

Author Response

The manuscript entitled "Quantitative monitoring of endo- and ectoparasites of domestic ruminants: a comparative review". Title, abstract and overall rationale of work to some extent is good. However, there are some major concerns, which needs to be addressed and needs substantial revision.

1) Introduction section is too lengthy and I would like to suggest author would be better to represent some information in the form of image(s).

Reply: we accepted the reviewer’s comment and reduced the length of the introduction section keeping only the fundamental parts of the text. Besides, the part of the introduction that was describing the differences among the groups of parasites is now moved to a new section (2. Methods, newly introduced as per suggestion of reviewer 3) and it’s supported by a figure (Figure 1), in order to reduce the text to the minimum.

2) I would like to suggest author, they must be include these ectoparasites such as fleas and ticks.

Reply: our decision to include only lice and mange mites followed the relevance of the host-based quantitative estimation to the monitoring and control approach of these parasites, such as for the different groups of endoparasites. On the contrary, fleas and ticks control need to consider the relevant role of the environmental phases of these parasites and the estimation of their burden on the host can only partially support the decision on the treatment. The justification for the exclusion of non-permanent parasites (including fleas and ticks) is now more explicitly presented in the new section 2. Methods (lines 128-139).

 3) I would like to suggest author to show a pictorial view of this (Sample size determination) part because this is too lengthy and not attractive.

Reply: we accepted the reviewer’s suggestion and created a new Figure (Figure 2) to show visually the sample size increment along with increasing farm size, according to the determination method proposed in the two different articles now cited in the corresponding text, which has been strongly reduced (lines 338-343).

 4) Author explain about the liver fluke parasite (Fasciola hepatica) and what’s about status of Schistosoma parasite in the animals.

Reply: We considered the parasites of the digestive and respiratory systems in our study, since these are the localizations of most of endoparasites whose abundance can be estimated through the quantification of the reproductive stages using fecal examination. Other internal organs can be affected by many different parasites whose life cycles differ greatly and whose importance is highly variable at a global level. The inclusion of all other endoparasites was therefore highly problematic. Fasciola can be considered part of the digestive system, whereas Schistosoma affects the circulatory system. This aspect is now better clarified by the new Figure 1, where endoparasites are presented as parasites of the digestive and respiratory systems.

 5) A flowchart should be added to the article to show the clear mechanism of this research.

Reply: we believe that the methodology followed in this research is now clearer in the revised manuscript, since we expanded its explanation in the newly introduced Method section. However, it’s not clear to us which kind of flowchart the reviewer is suggesting to include, since we did not perform a systematic review (which needs mandatorily a flowchart for the article selection process), but a normal review. We focused on few specific aspects (the three points at lines 118-119) and data were extracted accordingly from the reviewed papers. The type of data extracted can be clearly inferred from the 3 Tables.

 6) The most drawback of this review article are: written part is more, no figure and too much lengthy written, I would highly suggest the author they must include 2-3 figure and also I suggest the reduce the written part and try to express in the picture form.

Reply: we reduced the written test of the Introduction section, in agreement also with previous comments, and introduced two new figures to make it easier for the reader to understand the starting points of our study.

 7)  Author need to summarize main theme of this review and need to explain and highlight about the strength of this review.

Reply: as per reviewer’s suggestion, we expanded the Conclusion section to highlight the main findings of the review (lines 851-898).

 8) In this section (Bronchopulmonary nematodes, tapeworms and liver flukes) author explain about the different technique using to determine the parasite line number (525-536) however, there are several modern technique is are available to detect of these parasites live qRT-PCR, ELISA, Gene sequencing and now a day most of the lab using these techniques to identify the parasites.

Reply: in the new Method section it’s explained that we considered in our review only “parasitological methods based on the direct identification and count of adult, immature or reproductive stages of parasites and applicable in alive animals”, since these methods are still the more widespread and sufficiently homogeneous approaches in most of routine diagnostic laboratories. This is now better specified at lines 140-144 However, we mentioned in many points, in both Findings and Discussion sections, the new modern techniques, based on a molecular or serological approach. We are aware that these techniques may allow in the future a standardized quantitative approach to parasite burden estimation, however this is still not the case in most areas of the world and for many (neglected?) parasites.

 9) I am totally agree with the author they explain about the threshold definition section and I would like to suggest author also explain the limitation of this section.

Reply: we mentioned in many points of the Findings and Discussion sections (lines 568-571; 586-588; 776-782; 791-794) the limitation of the use of thresholds.

Reviewer 2 Report

The present study was based on a literature review aimed at the analysis of the characteristics of quantitative monitoring of parasites.
The abstract is confusing and does not provide much information.
The key words should be different from the title.
The tables are confusing.
DID NOT TALK ABOUT CLINICAL ASPECTS LIKE PHAMACHA AND BIOCHEMICAL AND MOLECULAR TECHNIQUES THAT COULD BE USED.
Inconsistent conclusions

Author Response

Reply: we tried to address part of the comments raised by reviewer 2. In particular some key words were changed and the conclusions section was also expanded, in agreement with reviewer 1 suggestion. Other comments of the reviewer 2 actually consist in opinions (e.g. “abstract/tables are confusing”) not supported by a clear indication of the weak points of the specific sections of our work and it’s therefore impossible for us to understand how to improve the manuscript.

Reviewer 3 Report

In their manuscript, Maurizio et al. summarize current approaches to the monitoring and quantification of Endo- and ectoparasites in domestic ruminants. The paper is highly relevant and generally very well written and I only have a few comments for improvement.

I missed a methods section stating how the literature search was conducted and based on what criteria articles were included or excluded.

Minor comments:

L. 19/20: I would suggest the present tense here instead of past tense.

L. 25: I think “episodes of” can be removed, it sounds as if resistance would be a transient phenomenon, which we know it isn’t.

L. 29: “Of the opposite”? This would mean that there is no evidence – but I think you mean that there is some preliminary evidence. Please rephrase.

L. 52: “depending also on…”

L. 57: “Each parasitic group has…”

L. 130: ”according to a newly…”

L. 162 ff.: Maybe it should be mentioned here that the unequal contribution of individual animals to FECs of pooled samples can impact the correlation.

Table 1: Please explain what “global” means in this context – all animals from the herd?

L. 251: “in ruminants”

L. 296 ff.: please set parasite genus names, such as Haemonchus, Oesophagostomum and Eimeria, in the following text in italics.

L. 311 ff.: Regarding liver flukes, intermittent shedding of parasite eggs due to storage in the gall bladder can impact FECs, this should be either mentioned here or in the discussion section.

L. 331: “semi-quantitatively defining…”

L. 375: “endoparasites”

L. 562: Delete “resulted.”

Author Response

In their manuscript, Maurizio et al. summarize current approaches to the monitoring and quantification of Endo- and ectoparasites in domestic ruminants. The paper is highly relevant and generally very well written and I only have a few comments for improvement.

I missed a methods section stating how the literature search was conducted and based on what criteria articles were included or excluded.

Reply: in agreement with the reviewer’s comment, we decided to insert a new section (2. Methods), to explain better the inclusion and exclusion criteria, although we would like to underline that our study was not a systematic review.

Minor comments:

  1. 19/20: I would suggest the present tense here instead of past tense.

Reply: for uniformity with the remaining text of the abstract, we left the verbs conjugated in the past tense.

  1. 25: I think “episodes of” can be removed, it sounds as if resistance would be a transient phenomenon, which we know it isn’t.

Reply: we agree with the reviewer and modified it accordingly.

  1. 29: “Of the opposite”? This would mean that there is no evidence – but I think you mean that there is some preliminary evidence. Please rephrase.

Reply: we meant that there is already some evidence that the treatment of endoparasites with endectocides can cause the onset of resistance in ectoparasites (while it is still not clear whether the treatment of ectoparasites can cause resistance in endoparasites). We modified the sentence to clarify it.

  1. 52: “depending also on…”

Reply: done.

  1. 57: “Each parasitic group has…”

Reply: done.

  1. 130: ”according to a newly…”

Reply: done.

  1. 162 ff.: Maybe it should be mentioned here that the unequal contribution of individual animals to FECs of pooled samples can impact the correlation.

Reply: we agree with the reviewer and mentioned it. The caption of Table 1 was also implemented to add this information for the included studies.

Table 1: Please explain what “global” means in this context – all animals from the herd?

Reply: we used the term “global” according to the definition used by the authors of the study we refer to (Rinaldi et al. 2019). “Global” means that the pool was made from all the sampled individuals in a farm. Since the sample size ranged from 9 to 20 animals per farm, the size of the “global” pools ranged accordingly. We added a note below Table 1 to clarify it.

  1. 251: “in ruminants”

Reply: done.

  1. 296 ff.: please set parasite genus names, such as Haemonchus, Oesophagostomum and Eimeria, in the following text in italics.

Reply: we modified them accordingly, also in the following lines.

  1. 311 ff.: Regarding liver flukes, intermittent shedding of parasite eggs due to storage in the gall bladder can impact FECs, this should be either mentioned here or in the discussion section.

Reply: we mentioned this aspect at lines 518-519, as per reviewer’s suggestion.

  1. 331: “semi-quantitatively defining…”

Reply: done.

  1. 375: “endoparasites”

Reply: done.

  1. 562: Delete “resulted.”

Reply: done.

Round 2

Reviewer 1 Report

The authors have addressed almost the concerns raised in the previous version of the manuscript and the quality has improved after incorporating required modifications. However, I am still not satisfied with figure 1 because this figure look like table and I suggest the author please make pictorial/picture format of this figure 1, after that, the manuscript will be considered for publication in this Journal.

Author Response

The authors have addressed almost the concerns raised in the previous version of the manuscript and the quality has improved after incorporating required modifications. However, I am still not satisfied with figure 1 because this figure look like table and I suggest the author please make pictorial/picture format of this figure 1, after that, the manuscript will be considered for publication in this Journal.

Reply: In consideration of the modification in the title (as per suggestion of reviewer 2), we decided to add also few sentences in the Methods section to clarify the criteria used to include or exclude (e.g. Schistosoma) the different groups of ecto- and endoparasites. Consequently, most of the information that were somehow displayed in the figure 1 are actually now present in the text, although in a synthesized way, compared to the first submission. In our opinion, in the present form of the manuscript, the figure 1 is not necessary and therefore we deleted it.

Reviewer 2 Report

Quantitative monitoring of endo- and ectoparasites of domestic 2 ruminants: a comparative review

Instead, temporary (mosquitoes, flies, sandflies) and periodic (ticks, dipterans causing 64 myasis, fleas) ectoparasites were excluded from our research, in consideration of the im- 65 portant role of the environment for their survival and transmission.

Therefore, the title cannot be that.

You are mentioning own work and in a restricted locality.

But in figure 1 it cites ticks. Moreover, it is incomplete on all organisms cited.

In the abstract should put the conclusion of the review.

In figure 2, it refers to which species of ruminant? It is not self explanatory. What is the period?

Tables  are nuclear

Little has been commented on biomolecular and other techniques such as phamacha.

I suggest only revising for endoparasites which was the main wording, but even so with additions

Author Response

Quantitative monitoring of endo- and ectoparasites of domestic 2 ruminants: a comparative review

Instead, temporary (mosquitoes, flies, sandflies) and periodic (ticks, dipterans causing 64 myasis, fleas) ectoparasites were excluded from our research, in consideration of the im- 65 portant role of the environment for their survival and transmission.

Therefore, the title cannot be that.

Reply: we accept Reviewer comment and modified accordingly the title into “Quantitative monitoring of selected groups of parasites in domestic ruminants: a comparative review”. Few parts of the main text were also modified to harmonize the manuscript to the new title (e.g. at the beginning of the Method section).

You are mentioning own work and in a restricted locality.

Reply: it’s not very clear which work the Reviewer refers to, however we mentioned the original methodologies used in three our studies since they are relevant to the review. The study area does not matter.

But in figure 1 it cites ticks. Moreover, it is incomplete on all organisms cited.

Reply: the aim of the figure was to summarize the main epidemiological features of different groups of parasites; therefore, it can hardly be complete. However, the figure has been deleted in the revised manuscript.

In the abstract should put the conclusion of the review.

Reply: we changed the last sentences in the abstract to include the main conclusions of the review.

In figure 2, it refers to which species of ruminant? It is not self explanatory. What is the period?

Reply: the figure presents a methodological proposal formulated in two different papers. The first paper refers to ruminants (in general), whereas in the second paper (Maurizio et al., 2021) the method was proposed for goats, although it could be easily extended to other ruminant species. We added a note to specify this point. The (sampling) period is not relevant to this issue.

Tables are nuclear

Reply: we suppose that the Reviewer is intending that the tables are “unclear”. Actually, we tried to display in a standardized way the complexity of the topics, considering the high heterogeneity of the literature. Other reviewers did not comment on the tables, and only reviewer 3 suggested minor corrections to improve the clarity of Table 1, already implemented in the first revision.

Little has been commented on biomolecular and other techniques such as phamacha.

Reply: These methodologies were excluded from the review, as clearly specified in the newly introduced Method section. However, we added a sentence on Famacha (lines 519-521) and we already mentioned biomolecular techniques in different parts of the original submission for a matter of comparison, integration or future perspective.  

I suggest only revising for endoparasites which was the main wording, but even so with additions

We understand that the argument we decided to tackle is wide and complex, however the comparison between endo and ecto-parasites quantitative monitoring approaches was the real starting point of our work, and we believe that it’s one of the main points of novelty of our study. In our opinion, some findings of the review regarding permanent ectoparasites hold interest for the scientific community. Consequently, we believe that focusing only on endoparasites will not improve the overall quality of the manuscript.